# Investigating Domain Gaps for Indoor 3D Object Detection

## Abstract

As a fundamental task for indoor scene understanding, 3D object detection has been extensively studied, and the accuracy on indoor point cloud data has been substantially improved. However, existing researches have been conducted on limited datasets, where the training and testing sets share the same distribution. In this paper, we consider the task of adapting indoor 3D object detectors from one dataset to another, presenting a first comprehensive benchmark with commonly used ScanNet and SUN RGB-D datasets, as well as our newly proposed large-scale SimRoom and SimHouse datasets by a 3D simulator with far greater number of objects and more precise annotations. Since indoor point cloud datasets are collected and constructed in different ways, the object detectors are likely to overfit to specific factors within each dataset, such as point cloud quality, room layout configuration, style and object size. We conduct experiments across datasets on different adaptation scenarios, analyzing the impact of different domain gaps on 3D object detectors. We observe that through our evaluated domain gap factors, synthetic-to-real adaptation is the most difficult adaptation hurdle to overcome. We also introduce several domain adaptation approaches to improve adaptation performances, providing a first baseline for domain adaptive indoor 3D object detection, hoping that future works may propose detectors with stronger generalization ability across domains.

## 1 Introduction

As a fundamental task for 3D perception and indoor scene understanding, indoor 3D object detection has been extensively studied. Most 3D object detectors for indoor scenes are designed for point cloud data due to its adaptability, precision, and richness of information. Indoor 3D detectors such as voting-based VoteNet by Qi et al. (2019) and transformer-based Pointformer by Pan et al. (2021) and V-DETR by Shen et al. (2023) have achieved remarkable success in identifying and localizing objects in point clouds.

Though achieving great progress, existing indoor 3D object detection researches are mainly conducted on ScanNet dataset by Dai et al. (2017) and SUN RGB-D dataset by Song et al. (2015). Detector are trained and evaluated within each dataset where the training and testing set share the same data distribution. As illustrated in Figure 1(a), point clouds in ScanNet dataset are constructed by thorough scans of RGB-D videos, thus with higher quality but relatively fewer scenes. Point clouds in SUN RGB-D dataset are converted through single RGB-D images, exhibiting large areas of point omission. In practical applications, detectors trained on data from a specific domain may need to generalize to deployment environments with distributional shift, i.e. domain adaptation problem which has not yet been explored in indoor 3D object detection field.

Domain adaptation for point cloud data has been studied on object classification task Qin et al. (2019); Shen et al. (2022); Cardace et al. (2023), outdoor LiDAR detection task Wang et al. (2020); Liu et al. (2024); Chang et al. (2024) and indoor semantic segmentation task Ding et al. (2022). However, object detection on indoor point clouds has the following challenges: (1) Indoor point clouds can be constructed through multiple ways, creating more aspects of domain gap factors with larger domain gap to be solved. (2) Indoor 3D object detection requires to distinguish multiple object categories (15 categories in our proposed benchmarks) with higher object diversity, resulting in richer semantics to be learned and adapted. (3) Localizing objects in indoor 3D scenes faces the challenges of close

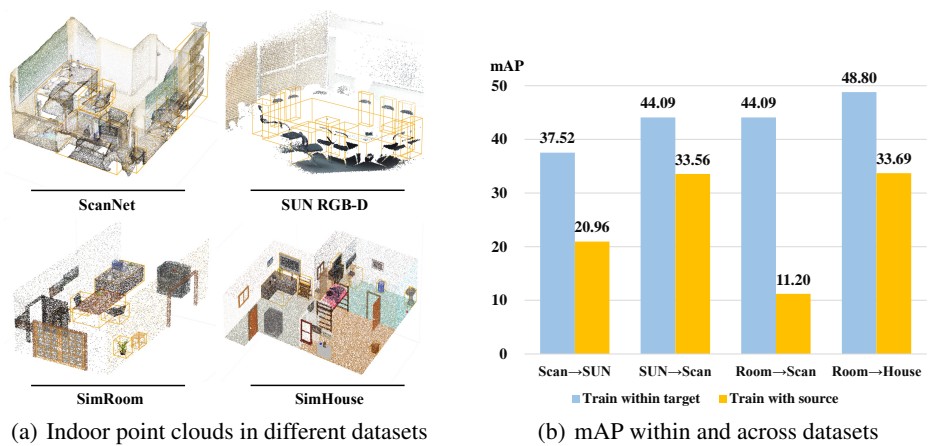

(a) Indoor point clouds in different datasets      (b) mAP within and across datasets

Figure 1: (a) Indoor point cloud samples in different datasets. Point clouds in different domains exhibit significant differences in point cloud quality, room layout, sytle and so on. (b) mAP within and across datasets. "Scan", "SUN", "Room" and "House" represents ScanNet, SUN RGB-D, SimRoom and SimHouse respectively. The performance shows a drastic decline in cross-dataset evaluation.

proximity of objects, sometimes even overlapping bounding boxes (e.g. a chair placed under a table), which may not appear in classification, segmentation and outdoor LiDAR detection tasks.

To investigate different factors of domain gaps that impact the performance of indoor 3D object detectors, we propose large-scale simulated indoor scene datasets SimRoom and SimHouse as a complement of existing indoor 3D object detection datasets, and re-arrange the label space of different datasets to construct a benchmark for domain adaptation. As illustrated in Figure 1(a), our proposed SimRoom contains scenes of single rooms which is closer to ScanNet and SUN RGB-D layouts, while SimHouse scenes are combinations of multiple rooms. Both datasets are constructed by directly sampling points from 3D meshes created by a 3D simulator, thus contain an order of magnitude more objects than ScanNet and SUN RGB-D datasets, and more precise bounding box and category annotations. Furthermore, our proposed datasets do not incur costs for manual construction and labeling, offering better scalability.

Shown in Figure 1(a), comparing the four datasets, ScanNet has relatively high quality point clouds, SUN RGB-D exhibit obvious point omission, SimRoom has distinct synthetic style with single-room layout and SimHouse has multi-room combination. After re-arranging the label space to select 15 common categories, to combine the four datasets to form a series of domain adaptation benchmarks including: (1) ScanNet→SUN RGB-D: high quality to low quality point clouds. (2) SUN RGB-D→SUN RGB-D: low quality to high quality point clouds. (3) SimRoom→ScanNet: synthetic style to real rooms. (4) SimRoom→SimHouse: single-room to multi-room configuration.

Under such benchmarks, we conduct experiments on the commonly used detectors Qi et al. (2019); Pan et al. (2021); Shen et al. (2023). As shown in Figure 1(b), the VoteNet detector performs well within each dataset (blue bars, trained and evaluated within target dataset), but cross-dataset evaluation shows a significant performance drop (yellow bars, trained by source and evaluated on target dataset). Within each adaptation scenario, we control the domain gap factors and further analyze the challenge of adaptation hurdles including point cloud quality, room layout configuration, style and object size. We observe that each factor causes a significant drop in detection performance, and among all these domain gap factors, synthetic-to-real gap is the most challenging one. Even with far greater number of training instances, synthetic-to-real setting shows the most severe performance degradation. We also implement several commonly used domain adaptation approaches in other tasks to improve adaptation performances. These approaches serve as a first baseline for domain adaptive indoor 3D object detection, hoping that future works may propose detectors or frameworks with stronger generalization ability across domains.

We summarize the contributions of this paper as follow:

- We propose the first domain adaptation series benchmarks for indoor 3D object detection with ScanNet, SUN RGB-D, and our newly proposed SimRoom and SimHouse datasets, proposing high-to-low-quality, low-to-high-quality, synthetic-to-real and single-to-multi-room adaptation scenarios.

- We conduct extensive experiments to investigate the challenges of multiple domain gap factors, and observe that synthetic-to-real adaptation is the most challenging factor.

- We implement several domain adaptation approaches to improve the performance of indoor 3D detectors, providing a first baseline for domain adaptive indoor 3D object detection.

## 2 RELATED WORKS

### 2.1 INDOOR POINT CLOUD DATASETS

As a universal 3D structure representation, point cloud data has been widely used in 3D indoor scene understanding tasks. Various 3D indoor scene datasets Armeni et al. (2016); Song et al. (2015); Dai et al. (2017) have been proposed by multiple construction ways. Song et al. (2015) propose SUN RGB-D dataset by converting single RGB-D images into 3D point clouds. Dai et al. (2017) propose ScanNet dataset by collecting RGB-D videos through large amount of indoor rooms with rich semantic annotations, and has been the most widely used dataset in indoor 3D object detection and semantic segmentation task. Simulated datasets like Structured3D Zheng et al. (2020) and 3D-Front Fu et al. (2021) are proposed to accomplish large-scale training with relatively low cost. However, these datasets are created by human designers and thus still have high construction cost. Deitke et al. (2022) offers an indoor structure generation framework with a fully simulated dataset which mainly focus on embodied AI tasks such as navigation and rearrangement.

Though helping downstream tasks to achieve great success, these datasets and benchmarks still have limitations in terms of data scale, data quality, and label space. Moreover, these benchmarks all assume that the training set and testing set share the same distribution, while in these paper we focus on adapting the indoor 3D detector across domains. We propose SimRoom and SimHouse datasets by the 3D synthetic framework proposed by Deitke et al. (2022), and combining with ScanNet and SUN RGB-D datasets to propose the first domain adaptive indoor 3D object detection benchmark.

### 2.2 INDOOR 3D OBJECT DETECTION ON POINT CLOUDS

Indoor 3D object detection is a fundamental task in indoor scene understanding, serving as the upstream task for 3D visual grounding, question answering and navigation. 3D-SIS by Hou et al. (2019) back projects the 2D feature vector onto the associated voxel in the 3D grid to achieve 3D detection and segmentation. Qi et al. (2018) frustum the PointNet framework by Qi et al. (2017) for 3D object detection. VoteNet by Qi et al. (2019) adopt deep hough voting strategy to train the model to group points for detection proposal generation. In recent years, transformer architectures are also introduced by Shen et al. (2023); Yang et al. (2023); Zhu et al. (2023).

Though achieving great success, these studies all focus on performances where the training and testing set share the same data distribution. The domain adaptation ability of indoor 3D object detectors has not yet been explored. In this paper, we propose the first domain adaptation benchmark, and mainly conduct experiments on VoteNet by Qi et al. (2019) due to its popular use among indoor scene understanding tasks and its flexible and lightweight architecture.

### 2.3 DOMAIN ADAPTATION FOR POINT CLOUDS

Domain adaptation for point clouds has been studied on object classification, indoor semantic segmentation and LiDAR point cloud detection tasks.

Qin et al. (2019) first propose a domain adaptation approach for object classification by reconstructing local points. Following the pioneer work, Shen et al. (2022); Cardace et al. (2023) propose global feature reconstruction and self-training methods to improve domain adaptation performance for 3D classifiers. Ding et al. (2022) studies domain adaptation for indoor semantic segmentation by synthetic-to-real data augmentation and mixing. However, domain adaptation approaches for classification and segmentation cannot be fully transferred to object detection task. We first construct

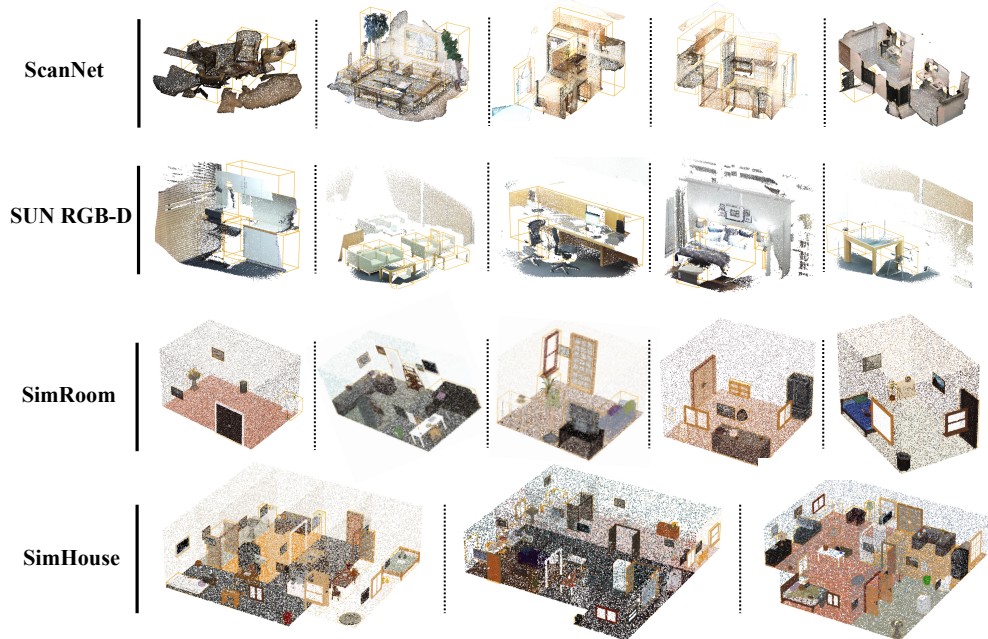

Figure 2: Visualization for typical scenes in the selected datasets of our proposed benchmarks.

domain adaptation benchmarks for indoor 3D object detection, and implement part of the commonly used adaptation techniques as a first baseline.

In LiDAR detection in autonomous driving scenes, Wang et al. (2020) first propose to test the LiDAR point cloud detector across multiple autonomous driving datasets. Following Wang et al. (2020), several studies Hu et al. (2023); Liu et al. (2024); Chang et al. (2024) explore domain adaptation approaches including mean teacher self-training, point completion and pseudo label refinement to improve the generalization ability of LiDAR point cloud detectors. However, domain adaptation for LiDAR detection in autonomous driving scenarios differs significantly from indoor scenes, with the latter presenting unique challenges: (1) Outdoor point clouds are all constructed by LiDARs, while indoor point clouds are constructed through far more different ways, resulting in multiple domain gap factors with more complicated domain gaps. (2) Outdoor domain adaptation focus only on cars Wang et al. (2020); Yang et al. (2021), while indoor object detection task requires to identify and locate far more object categories (15 in our benchmarks) with rich semantic information. (3) Distance between cars are relatively large, while indoor scenes exhibit closely placed object with even overlapped bounding boxes. In this paper, we investigate the domain gaps for indoor 3D object detection.

## 3 DATASETS AND BENCHMARKS

To build the domain adaptation benchmarks for indoor 3D object detection, in this section, we review existing indoor 3D object detection datasets in Section 3.1 and introduce our newly proposed datasets SimRoom and SimHouse in Section 3.2. We then analyze the differences of datasets in multiple aspects, and introduce our proposed domain adaptation benchmarks in Section 3.3.

### 3.1 EXISTING INDOOR 3D OBJECT DETECTION DATASETS

**SUN RGB-D** SUN RGB-D dataset by Song et al. (2015) is constructed by 10,335 single RGB-D images captured by 4 different sensors. It also provide tools to convert single RGB-D frames to point clouds. The original split use 5,073 scenes as training set and 4,828 for testing. Instance-level object oriented bounding box annotations with 620 fine-grained object categories are provided for 3D object detection and orientation estimation task.

**ScanNet and ScanNet++** ScanNet dataset by Dai et al. (2017) is one of the most commonly used dataset in indoor 3D scene understanding. By thoroughly scanning real-world indoor rooms, Dai et al. (2017) collect 2.5M frames of RGB-D videos and reconstruct them into CAD models which can be converted into point clouds. The original split use 1,201 scenes as training set and 312 scenes for testing, containing 529 fine-grained object categories. ScanNet++ dataset by Yeshwanth et al. (2023) has been recently proposed as a complement of ScanNet. It captures higher fidelity indoor scenes by professional devices, containing 230 scenes for training and 50 for testing.

Due to the massive human labor of 3D data collection and calibration, the scale of indoor 3D datasets is limited. Additionally, human annotated 3D datasets unavoidably contain label noise due to the challenge of point cloud annotation, which has been evaluated by Yu et al. (2024). Moreover, the real-world datasets lack extensibility, insufficient to investigate multiple aspects of domain gap that may occur in practical scenarios.

### 3.2 OUR PROPOSED SIMROOM AND SIMHOUSE

To address the data scale issue, researches have tried to use simulators to create large-scale datasets with precise annotations at a low cost such as Zheng et al. (2020); Fu et al. (2021), but they mainly focus on room layout estimation, generation or embodied AI tasks. Moreover, the human designed datasets still faces the high cost of construction and can hardly be scaled up.

In this paper, to fill the void in synthetic indoor 3D object detection datasets, and to investigate the domain gap of synthetic and real scenes, we use the 3D simulator proposed by Deitke et al. (2022) to generate SimRoom and SimHouse datasets. Specifically, we randomly sample large-scale diverse house layouts following the ProcTHOR procedure of Deitke et al. (2022), including room spec generation, floorplan generation, adding structure materials, large object, wall object and finally surface objects. In this procedure, the precise location and surface information of room structures and objects can be obtained, without the need of human labor. We then export the layout information into mesh format (e.g. .glb files) with object shapes and surface information, and uniformly sample points from the mesh surface to construct point clouds.

The ProcTHOR framework is originally designed for embodied AI tasks such as navigation, thus generates combination of rooms to form a house, while in indoor 3D object detection task, most datasets are captured in single rooms. To control this factor and investigate the influence of room configuration, we split single rooms from a part of the aforementioned house point clouds. For a selected house scene, we use the room information in generation process to collect all the 3D assets in each room, and convert these room sub-meshes into independent point clouds. We name the split single room point clouds as SimRoom, and the originally generated house point cloud as SimHouse. In practice, we totally generate 10,000 house scenes, in which 7,306 scenes are randomly selected to construct SimHouse dataset, and 2,000 houses are further split into 7,202 single rooms. In both SimRoom and SimHouse datasets, we randomly split 6,000 point cloud scenes as training set, and the rest of the scenes as testing set. The scene number of both SimRoom and SimHouse is more than 5 times larger than ScanNet, and the number of bounding box annotations is an order of magnitude larger than ScanNet and SUN RGB-D.

Note that we choose to use a generative framework rather than manually designing simulated data scenarios, which also offers advantages in scalability and controllability. On one hand, the generative framework allows the generation of data at any scale, facilitating the study of the scale-up characteristics of 3D models. On the other hand, using this framework enables flexible control over the generated indoor 3D scene configurations, including object layout combinations and the semantics of the objects used, allowing for more detailed control-variable research of the domain gap. Relevant experiments will be discussed in the following sections.

### 3.3 DOMAIN ADAPTATION BENCHMARKS

#### 3.3.1 ANALYSIS OF DATASET DIFFERENCES

In Figure 2 we present typical scenes of each dataset, and in Figure 3(a) we show single object samples in each dataset. In Table 1 we show the static information of the datasets. We analyze the differences of these datasets as follows:

Table 1: Comparison of the indoor 3D object detection datasets

| Dataset | Training / Testing scenes | Construction method | Object number | Multi-rooms |
|---------|---------------------------|---------------------|---------------|-------------|
| SUN RGB-D | 5,073 / 4,828 | Single RGB-D images | 54,376 | ✗ |
| ScanNet | 1,201 / 312 | RGB-D videos | 36,759 | ✗ |
| ScanNet++ | 230 / 50 | RGB-D videos | 18,959 | ✗ |
| SimRoom | 6,000 / 1,202 | Synthetic meshes | 176,869 | ✗ |
| SimHouse | 6,000 / 1,306 | Synthetic meshes | 686,211 | ✓ |

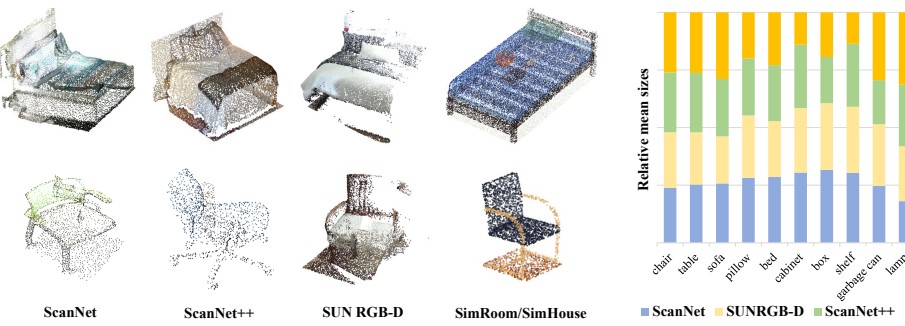

(a) Examples of objects in different datasets  (b) Object mean sizes in different datasets

Figure 3: (a) Examples of objects in different datasets. The datasets exhibit differences in point cloud quality and style. (b) Object mean sizes (relative mean volume) in different datasets. Certain categories show significant size differences across datasets.

**Data scale**: As shown in Table 1, ScanNet dataset has a relatively small scale of totally around 1.5k scenes and 37k annotated objects. The high fidelity ScanNet++ dataset could only provide hundreds of scenes and around 19k objects. SUN RGB-D dataset is able to provide over 54k scenes. But limited by its quality, the annotation of object becomes difficult, providing only around 37k bounding boxes. Our proposed synthetic datasets have the advantages of low cost and precise annotation, thus able to offer around 7k high quality scenes with over 176k/686k precise bounding box annotations of objects which is an order of magnitude more than real-world datasets.

**Point cloud quality**: As can be visualized in Figure 2 and Figure 3(a), thoroughly scanned ScanNet dataset have high quality of point clouds representing the real furniture. SUN RGB-D point clouds exhibit obvious point omission since they are converted from single RGB-D images. Our proposed synthetic datasets have high quality point clouds without the problem of scan omission.

**Room layout configuration**: As shown in Table 1, existing 3D object detection datasets are mainly single room scenes, while in downstream embodied AI tasks, detecting objects in complex environments is required. Our proposed SimHouse provides house scenes with multi-rooms and object annotations to study the impact of layout configuration domain gap.

**Style(synthetic to real)**: As illustrated in Figure 2 and Figure 3(a), the style difference exists between synthetic and real-world datasets. Though providing high quality point clouds, synthetic datasets lack the texture details and material realism of objects in real-world environments. This gap presents a challenge for adapting it to real-world scenarios.

**Object information**: Objects in different datasets exhibit two types of differences. From a global perspective, there are significant differences in the mean size of objects. From an individual perspective, different datasets may contain objects with unseen semantic information. In Figure 3(b) we present the relative mean volumes of each object category. Four equal length of colors of bars represent that the mean size of the object is similar across the four datasets. In the contrary, certain categories such as "cabinet", "garbage can" and "toilet" show significant size differences across datasets. The impact of object size difference will be further discussed in Section 4.3. As shown in Figure 3(a), furniture of different forms may appear in different datasets, testing the model's cross-domain generalization ability in terms of semantics. We conduct experiments in Section 4.3 to evaluate the impact of this domain gap.

### 3.3.2 RE-ARRANGING LABEL SPACE

Conventional domain adaptation requires different domains to share the same label space, i.e. target categories in object detection. (Partial domain adaptation or open vocabulary domain adaptation is our of the scope of this work.) The original annotations in the aforementioned datasets contains hundreds of fine-grained object categories with extreme long-tail distribution. Song et al. (2015) evaluate the performance of 10 most common categories on SUN RGB-D, and ScanNet 3D object detection report the results on 18 categories started from Hou et al. (2019). The direct subset of these benchmarks is too limited to fully utilize the numerous instance annotations available in the datasets.

We cataloged the original fine-grained object categories and the number of objects in each category within each dataset. After manually checking and eliminating typos in the category names, we merged the fine-grained categories into broader categories. (For example, "dining table" and "office table" are both merged into "table" category.) In the merged broader categories, we select ones that satisfy the following conditions: (1) The category is an object instead of room structure such as "wall" or "window". (2) The category each contains at least 150 objects. In this way, we select 15 object categories shown in the horizontal axis of Figure 3(b) as the unified label space in our domain adaptation benchmarks.

### 3.3.3 DOMAIN ADAPTATION SCENARIOS

Based on the analysis of differences between datasets, we propose 4 domain adaptation settings based on the practical requirements:

**High quality to low quality point clouds**: Detectors trained on thoroughly scanned datasets should be able to adapt to environments where the point clouds are of lower quality. Under this setting, we use ScanNet as source domain and SUN RGB-D as target domain.

**Low quality to high quality point clouds**: In contrast with fully to partial scan, in some practical scenarios where training data faces the challenge of point omission, the detector is required to adapt to normal scenes where the point clouds are of higher quality. Under this setting, we use SUN RGB-D as source domain and ScanNet as target domain.

**Synthetic to real adaptation**: Fully leveraging the large-scale, high-quality and precisely annotated synthetic data, the detectors should be capable of adapting to real-world scenarios to meet practical requirements. Under this setting, we use SimRoom as source domain and ScanNet as target domain.

**Single-room to multi-room adaptation**: Downstream tasks such as navigation requires the model to understand complex scenes, i.e. detecting objects in the house scenarios. Adaptation from single-room training data to multi-room environments is a challenging and unique task in indoor 3D object detection. SimHouse dataset has room layout configuration gap between all other datasets of single rooms. To control other domain gap factors, we focus on SimRoom→SimHouse adaptation in which both datasets are constructed from synthetic meshes.

Note that the data scale of ScanNet++ dataset is far smaller that the other datasets. Through experimental analysis on data scale factor, we do not include ScanNet++ dataset into our proposed domain adaptation benchmarks, which will be elaborated in Section 4.2.

## 4 EXPERIMENTS AND ANALYSIS

### 4.1 SETUP

In our experiments, we apply commonly used indoor 3D object detector VoteNet by Qi et al. (2019) to evaluate the cross domain performance and domain adaptation approaches. We also conduct experiments on transformer-based Pointformer by Pan et al. (2021) and V-DETR by Shen et al. (2023), which can be seen in supplementary materials. We randomly sample 40,000 points per scene for training. In addition to XYZ coordinates, we include height feature and RGB color feature for each point. We use data augmentations including random flipping, random scale, brightness and color shifting, color jitting and random color dropping. We train the network end-to-end with AdamW optimizer and batch size 64 for 90 epochs. The initial learning rate is 0.008, and is decreased by $10\times$ after 65 and 80 epochs.

Table 2: Detailed results on our proposed benchmarks. "scn2sun", "sun2scn", "rm2scn" and "rm2hse" represents ScanNet→SUN RGB-D, SUN RGB-D→ScanNet, SimRoom→SimHouse and SimHouse→SimRoom respectively. "src", "MT" and "tgt" denotes source only training, mean teacher approach and target domain training respectively.

| | | chair | table | sofa | pillow | bed | cabinet | box | shelf | gbg.can | lamp | sink | dresser | tv | toilet | plant | mAP |
|---|---|---|---|---|---|---|---|---|---|---|---|---|---|---|---|---|---|
| scn2sun | src | 36.6 | 38.7 | 32.3 | 1.4 | 55.8 | 2.2 | 0.2 | 15.6 | 19.3 | 11.6 | 20.0 | 2.5 | 6.4 | 70.2 | 1.7 | 21.0 |
| | tgt | 70.3 | 55.8 | 75.6 | 4.6 | 80.1 | 5.7 | 0.8 | 31.3 | 42.3 | 26.4 | 38.2 | 22.1 | 13.5 | 90.6 | 5.6 | 37.5 |
| sun2scn | src | 74.3 | 54.3 | 56.4 | 2.6 | 61.7 | 10.2 | 2.6 | 41.2 | 27.4 | 24.4 | 22.3 | 19.8 | 25.5 | 74.0 | 6.7 | 33.6 |
| | tgt | 78.2 | 62.8 | 81.7 | 8.8 | 84.6 | 26.6 | 9.6 | 42.1 | 41.7 | 34.5 | 42.5 | 23.3 | 30.5 | 88.1 | 6.5 | 44.1 |
| rm2scn | src | 47.6 | 35.3 | 39.1 | 0.1 | 6.0 | 0.5 | 1.2 | 4.6 | 4.8 | 7.2 | 0.2 | 0.2 | 1.3 | 19.7 | 0.4 | 11.2 |
| | tgt | 78.2 | 62.8 | 81.7 | 8.8 | 84.6 | 26.6 | 9.6 | 42.1 | 41.7 | 34.5 | 42.5 | 23.3 | 30.5 | 88.1 | 6.5 | 44.1 |
| rm2hse | src | 35.5 | 50.6 | 60.9 | 4.7 | 47.2 | 9.0 | 13.2 | 6.8 | 48.4 | 27.0 | 31.6 | 29.4 | 45.9 | 57.9 | 37.5 | 33.7 |
| | tgt | 55.1 | 64.9 | 73.8 | 4.8 | 71.4 | 10.8 | 27.0 | 12.8 | 71.8 | 40.6 | 51.1 | 43.1 | 67.1 | 86.1 | 48.7 | 48.8 |

During inference, we post-process the proposals with dropping empty boxes (less than 5 points in a box), dropping too little boxes (boxes with side length less than 0.001 m) and 3D NMS (non-maximum suppression) with IoU threshold of 0.25. We report the evaluation metric of mAP (mean average precision) following the protocal of Song and Xiao (2016) with IoU threshold 0.25.

## 4.2 ANALYSIS OF DOMAIN GAPS

We present the basic experimental results on our proposed benchmarks in Table 2 and further domain adaptation results in Table 3 to analysis the impact of multiple domain gap factors mentioned in Section 3.3.1. We specially analyze the influence of data scale factor in domain adaptation in Table 5.

**Point cloud quality**: ScanNet→SUN RGB-D and SUN RGB-D→ScanNet benchmarks exhibit the domain gap of point cloud quality. Illustrated in Table 2 "scn2sun", training by ScanNet performs 20.96 mAP on SUN RGB-D evaluation set (line "src"), much lower than 37.52 mAP within target domain (line "tgt"). Conversely, the mAP of training by SUN RGB-D is also much lower than training by ScanNet when evaluated on ScanNet (Table 2 "sun2scn"), indicating the challenge of adaptation from both high to low and low to high quality datasets.

**Room layout configuration**: As shown in Table 2 "rm2hse", SimRoom trained model performs 33.69 mAP (line "src") which is much lower than the result within target dataset (48.80 mAP in line "tgt"), confirming the challenge of adapting single-room object detector to multi-room environments.

**Style(synthetic to real)**: SimRoom→ScanNet exhibit synthetic to real style gap. As illustrated in Table 2 "rm2scn", ScanNet trained detector performs well on its own evaluation set (44.09 mAP), but drastically decline to only 11.20 mAP (line "src") when trained on SimRoom, showing a huge performance gap. Compared with "sun2scn" where ScanNet also serve as target domain, with more data, the detector trained by SimRoom performs far worse than trained by SUN RGB-D. In "rm2hse" where SimRoom also serve as source domain, the "Souece only" performance is acceptable as 20.96 mAP. Moreover, as shown in Table 5, trained by only 230 scenes of ScanNet++ dataset performs better than by 6,000 scenes of SimRoom. Qualitative visualization results in Figure 4(a) also shows that model trained with SimRoom struggle to predict the right boxes in ScanNet validation set. In conclusion, synthetic-to-real gap is the most challenging hurdle for domain adaptive indoor 3D object detection among our proposed benchmarks.

In Table 4 we further analyze the detailed domain gap of synthetic and real data. Leveraging the flexibility of the generative data framework, we integrated objects from the ScanNet dataset to generate object placement layouts, with the objects being in an intermediate state derived from real-world data. As shown in the table, incorporating real objects leads to a noticeable performance improvement compared to fully simulated data, highlighting the semantic domain gap between datasets. However, compared to the target domain Oracle, the performance gap remains significant, suggesting that differences in object placement layouts are a more critical domain gap factor.

Table 3: Results on our proposed benchmarks. "Scan", "SUN", "Room" and "House" represents ScanNet, SUN RGB-D, SimRoom and SimHouse datasets respectively.

| | Method | Scan→SUN | SUN→Scan | Room→Scan | Room→House |
|---|---|---|---|---|---|
| | Source only | 20.96 | 33.56 | 11.20 | 33.69 |
| Prior | Few shot(10) | 27.52 | 36.21 | 15.31 | 34.56 |
| | Few shot(100) | **29.23** | **41.95** | **30.04** | **38.36** |
| | Size prior | 21.73 | 33.95 | 12.95 | 33.30 |
| UDA | MT Tarvainen and Valpola (2017) | 28.49 | 35.14 | 11.41 | 33.64 |
| | VSS Ding et al. (2022) | 20.69 | 30.12 | **13.22** | 31.84 |
| | PPFA Zhang et al. (2024) | 23.52 | 34.77 | 12.19 | **36.33** |
| | RV Fan et al. (2022) | **29.12** | 35.61 | 12.35 | 33.74 |
| | OHDA Yunsong et al. (2024) | 29.07 | **36.03** | 12.02 | 33.55 |
| | Oracle(target) | 37.52 | 44.09 | 44.09 | 48.80 |

Table 4: Analysis of the impact of object semantic.

| Training data | SimRoom | | | SimRoom w/ ScanNet objects | | |
|---|---|---|---|---|---|---|
| Method | Source | RV | Oracle | Source | RV | Oracle |
| mAP on ScanNet | 11.20 | 12.35 | 44.09 | 18.76 | 20.64 | 44.09 |

**Excluding the factor of data scale**. The performance of deep neural networks rely on the scale of training data. Different datasets vary in data scale as shown in Table 1, the influence of which should be excluded from the analysis of domain gaps. ScanNet++ dataset offers only 230 scenes of training data which is far less than other datasets. Thus, we align the data scale in terms of scene number and report the results evaluated on ScanNet dataset, shown in Table 5. With only 230 scenes training data, but without domain gap, ScanNet training set achieves 22.54 mAP, significantly exceed 8.07 by ScanNet++, but also has considerable disparity with the whole dataset trained result 44.09. This indicates that the drastic performance decline of ScanNet++→ScanNet comes from not only domain gap, but also the data scale. Thus, we consider ScanNet++ unsuitable for studying domain adaptation in indoor 3D object detection, and exclude it from our proposed benchmarks. We also conduct experiments of data scale control on SUN RGB-D and SimRoom dataset. When only providing 230 scenes, SUN RGB-D training performs worse than ScanNet++, with SimRoom even worse. With 400 scenes used for training, SUN RGB-D could surpass ScanNet++. But even with 3,000 scenes, SimRoom could only reach 5.25 mAP, still lower than ScanNet++. These observations further confirm that the point omission domain of SUN RGB-D is a larger gap than that between ScanNet++ and ScanNet, and the synthetic-to-real gap is the most challenging.

### 4.3 DOMAIN ADAPTATION APPROACHES

We introduce several domain adaptation approaches including methods using target domain priors and unsupervised domain adaptation techniques to improve adaptation performance, presenting a first baseline for this problem.

#### 4.3.1 USING TARGET DOMAIN PRIORS

**Few-shot fine-tuning**: In Table 3 we report the result of fine-tuned with 10 and 100 labeled target scenes. All baselines show an improvement under 10-shot fine-tuning compared with the weak source only baseline. In SUN RGB-D→ScanNet, fine-tuned by 100 target scenes could reach a relatively high performance compared with oracle. Figure 4(b) further shows the performance by fine-tuning with different number of scenes. For every benchmark, tuning with 200 labeled target domain scenes gets the comparable performance with target domain fully trained model.

**Using object size prior**: As has been shown in Figure 3(b), different datasets exhibit mean object size difference. By simply adopting target domain statistical prior of mean sizes of each category

Table 5: Analysis the impact of data scale. "Scan", "SUN", "Room" and "Plus" represents ScanNet, SUN RGB-D, SimRoom and ScanNet++ datasets respectively. The number in () denotes the number of selected training scenes, where the full ScanNet++ training set contains only 230 scenes.

| Training data | Scan(230) | Plus(230) | SUN(230) | SUN(400) | Room(230) | Room(3,000) |
|---|---|---|---|---|---|---|
| mAP on Scan | 22.54 | 8.07 | 5.50 | 8.42 | 2.26 | 5.25 |

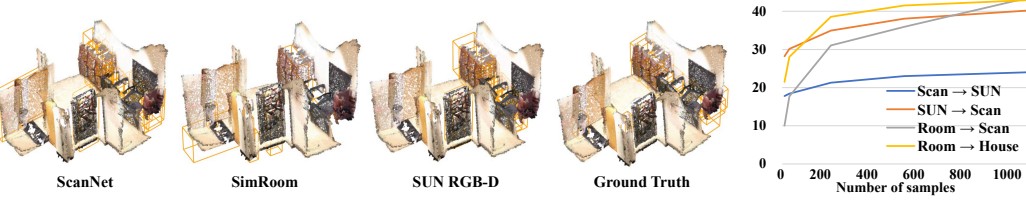

(a) Visualization of detection results in different domain adaptation settings  (b) Few-shot performance

Figure 4: Visualization of (a) Detection results on ScanNet dataset where the detector is trained on different datasets compared with ground truth; (b) Performance curve of few-shot fine-tuning.

instead of precise labels, reported in "Size prior", the detector shows performance gains generally. (SimRoom→SimHouse shows limited gain for the objects of the two datasets are sampled from the same pool, resulting in similar sizes.) This observation shed a light on future studies to estimate target domain object sizes.

### 4.3.2 UNSUPERVISED DOMAIN ADAPTATION

We evaluate different domain adaptation techniques on our proposed benchmarks, including classical framework MT Tarvainen and Valpola (2017), VSS Ding et al. (2022) for segmentation task, PPFA Zhang et al. (2024) and RV Fan et al. (2022) for classification task, and OHDA Yunsong et al. (2024) which has not been officially published yet. Note that due to the difference on task setting, we re-implement all the methods and make some adjustments to achieve better results on object detection task.

As shown in Table 3, mean teacher framework shows performance gains compared to source only results in each benchmark. In line "VSS", this high-to-low quality augmentation method works for ScanNet→SUN RGB-D and SimRoom→ScanNet benchmarks, for in these benchmarks higher quality point clouds. But in other benchmarks, this low-quality simulation method shows counter-productive. In line "PPFA", prototype-based feature alignment method with adversarial training and self-supervised learning techniques performs well in SimRoom→SimHouse benchmark, as the object-level distribution of the two datasets are similar, sharing a better category prototype. In line "RV" which is a unique method to improve classification consistency between different domains when generating pseudo labels, it performs well in most cases except in SimRoom→SimHouse, which indicates that RV can only improve the classification accuracy at the object level. However, the gap in SimRoom→SimHouse task is the object configuration and the number of rooms and the objects themselves are similar. This is why RV does not work well in this case.

## 5 CONCLUSION

To facilitate domain adaptation for indoor 3D object detection on point clouds, we propose the first benchmark combining the commonly used ScanNet and SUN RGB-D datasets, as well as our newly proposed large-scale synthetic dataset SimRoom and SimHouse. We propose 4 adaptation scenarios, and conduct extensive experiments to investigate the domain gap factors including point cloud quality, room layout configuration, style(synthetic to real) and object sizes. We observe that the synthetic-to-real adaptation gap is the most challenging hurdle among our evaluated ones. We also introduce domain adaptation approaches to improve the adaptation performances, proposing a first baseline. We hope future works could effectively address the domain gaps and make indoor 3D object detectors generalize well.

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
