# Supplementary Material for Investigating Domain Gaps for Indoor 3D Object Detection on Point Clouds

## A   Appendix

In this additional material, we offer more comprehensive information and analysis of our proposed benchmark. In Appendix A.1, we present the distribution of object categories in the datasets used in our benchmark. In Appendix A.2, we conduct experiments on other detectors. In Appendix A.3, we compare our proposed synthetic dataset with existing synthetic dataset. In Appendix A.4, we conduct more experiments to further analyze the domain gap factors and the results of domain adaptation approaches. In Appendix A.5, we report the computation resources required to reproduce our experiments, and list the limitations of this paper.

### A.1   Instance distribution of object categories

In this section, we summarize the object category distribution of our proposed SimRoom and SimHouse dataset, and compare the distribution with ScanNet by Dai et al. (2017) and SUN RGB-D by Song et al. (2015) which are used in our proposed domain adaptation benchmarks.

Figure 1 shows the object category distribution of different datasets. The number of objects in different categories in the ScanNet and SUN RGB-D datasets exhibits a significant long-tail distribution, with their overall trends being relatively similar. Although we considered the number of originally annotated objects in the datasets when selecting target categories, certain categories in ScanNet and SUN RGB-D, such as "plant" and "TV", contain only a small number of samples. This poses additional challenges for training the object detector. In contrast, the object category distribution in the SimRoom and SimHouse datasets differs significantly from that of the two real datasets. This is because the 3D simulator considered object diversity when generating scenes, resulting in relatively balanced scenes without a significant long-tail distribution. It is worth noting that despite the differences in distribution, our proposed SimRoom and SimHouse datasets ensure an absolute number (at least more than a thousand) of objects in each category. By using simulated datasets with a more balanced object category distribution and a greater absolute number of objects to train the detector, the issues of few samples and long-tail distribution can be effectively addressed. The focus of domain adaptation research will then shift to bridging the domain gap between the styles of simulated and real data.

### A.2   Performance on other detectors

We conduct experiment on newly proposed transformer-based Pointformer Pan et al. (2021) and V-DETR Shen et al. (2023). As shown inTable 1, the newly proposed detectors also face the performance drop when evaluated across datasets. We hope future works will propose general domain adaptation methods that works for detectors with multiple architectures.

### A.3   Results on other synthetic dataset

Several synthetic indoor datasets have been proposed for 3D layout generation or navigation tasks. They are mostly created by professional designers, making them expensive, non-configurable, and difficult to scale. As shown in Table 3, our proposed SimRoom and SimHouse datasets which are generated by ProcTHOR framework have following advantages: (1) 3D-Front dataset is created by professional indoor designers with fixed layouts, thus with high cost and could neither be extended

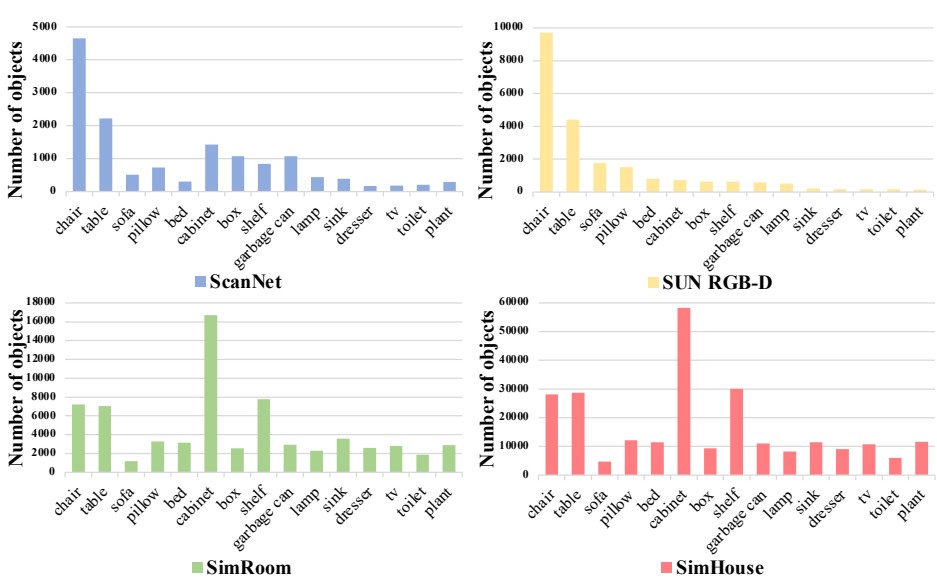

Figure 1: Object category distribution of different datasets in training set.

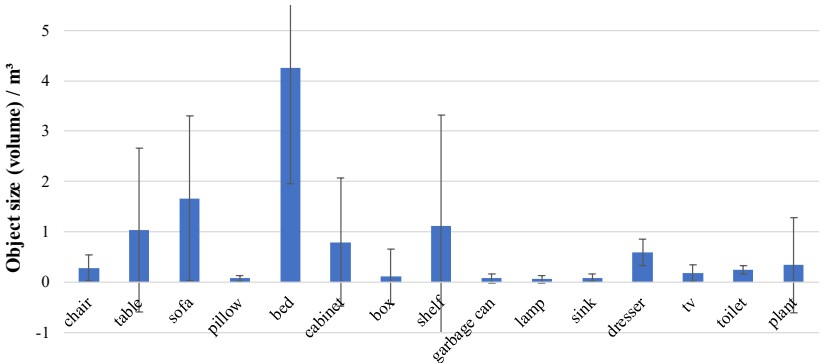

Figure 2: Object size mean and variance of each category in ScanNet.

Table 1: Results of Pointformer and V-DETR, trained on ScanNet.

| Test dataset | Pointformer Pan et al. (2021) | V-DETR Shen et al. (2023) |
|---|---|---|
| ScanNet | 45.82 | 49.30 |
| SUN RGB-D | 29.26 | 31.77 |

Table 2: Evaluation of SimRoom→SimHouse benchmark on subset of different room numbers.

| House number | 2 | 3 | 4 | 5 | 6 | 7 |
|---|---|---|---|---|---|---|
| Source only | 46.81 | 28.60 | 15.43 | 14.17 | 6.80 | 4.91 |
| Mean teacher | 39.07 | 27.11 | 17.16 | 14.64 | 7.45 | 5.17 |
| Oracle(target) | 64.97 | 53.39 | 49.33 | 47.95 | 37.79 | 37.75 |

nor easily modified. Our proposed datasets is extensible and could scale up to any size with very low cost. (2) 3D-Front dataset contains limited number of categories and instances. Some object categories in our benchmark which are common among indoor scenes such as TV, toilet, and garbage can are not included or annotated in 3D-Front. In ProcTHOR framework we could add 3D models of any category and could easily configure their numbers and layouts. We train the detector with 3D-Front dataset as source domain on 8 categories (our proposed benchmark include 15 categories). As shown in Table 4, the synthetic to real adaptation (3D-Front to ScanNet) remains challenging. Additionally, domain gap exists between manually designed layout and automatically generated layout (3D-Front to SimHouse).

We propose a part of our proposed datasets (validation set of SimRoom and SimHouse) as an example: link.

## A.4  FURTHER ANALYSIS ON DOMAIN GAPS

In this section, we conduct more experiments to further illustrate the domain gap factors including room configuration gap and object size diversity. Additionally, we further analyze the results of domain adaptation approach mean teacher.

### A.4.1  ANALYSIS OF ROOM CONFIGURATION GAP

In Table 2 we show the evaluation results of SimRoom→SimHouse benchmarks on subsets of SimHouse with different number of rooms. As the number of rooms in a house scene increases, the complexity of the scene layout rises, which inherently increases the detection difficulty within the target domain. This is evidenced by the decline in line "Oracle(target)" test results. The rising of room layout complexity significantly increases the adaptation difficulty for training on single-room SimRoom datasets. This is reflected in the more drastic decline in performance of source only trained model. For the mean teacher results, its performance is slightly lower than source-only on subsets with fewer rooms, but it improves in more complex scenes. This indicates that the mean teacher framework enables the model to better adapt to more complex environments. This result indicates that the domain gap in room layout configuration is an urgent issue to address. Our proposed SimRoom→SimHouse benchmark provides a platform to evaluate this adaptation capability. We hope that future domain adaptation efforts will generalize detectors to more complex real-world environments.

Table 3: Comparison of our proposed SimRoom / SimHouse and 3D-Front Fu et al. (2021) dataset.

| Dataset | Training / Testing scenes | Categories | Instance number | Multi-rooms | Extensible |
|---|---|---|---|---|---|
| SimRoom | 6,000 / 1,165 | 71 | 157,473 | ✗ | ✓ |
| SimHouse | 6,000 / 2,000 | 71 | 657,670 | ✓ | ✓ |
| 3D-Front | 5,000 / 1,813 | 26 | 149,228 | ✓ | ✗ |

Table 4: Results of training with 3D-Front dataset as source domain on VoteNet detector.

| Method | 3D-Front→ScanNet | 3D-Front→SimHouse |
|---|---|---|
| Source only | 13.21 | 21.18 |
| Oracle (target) | 47.96 | 37.78 |

Table 5: Evaluation on different subsets that vary in object size. "Larger gap" denotes the subset in which the objects' volumes are 2 times larger or smaller than the mean size in source domain. "Smaller gap" denotes the rest of the annotations which has a closer size with source domain.

| | SUN RGB-D→ScanNet | | SimRoom→ScanNet | |
|---|---|---|---|---|
| | Smaller gap | Larger gap | Smaller gap | Larger gap |
| Source only | 21.12 | 10.92 | 2.18 | 2.08 |
| Mean teacher | 20.82 | 12.92 | 5.45 | 2.44 |
| Oracle(target) | 32.48 | 13.52 | 32.48 | 13.52 |

A.4.2   ANALYSIS OF OBJECT SIZE DIVERSITY

To further illustrate the influence of object size, in SUN RGB-D→ScanNet and SimRoom→ScanNet benchmark we split the annotations of target evaluation set into subsets based on the object size. Specifically, to evaluate the influence of source domain object mean size, we split the evaluation annotation into 2 groups, filtering out the objects whose size(volume) is 2 times larger or smaller than the mean size of source domain. As shown in Table 5, in both benchmarks, the test results for groups with a larger gap in object size are significantly lower than for groups with more similar sizes. This indicates that object size has a direct and important impact on detector adaptation. In future work, estimating the average object size in the target domain and addressing the detection issues for objects with significant size differences from the source domain will be crucial for improving domain adaptation performance.

A.4.3   FURTHER ANALYSIS ON DOMAIN ADAPTATION APPROACH

We test the result of basic domain adaptation approach mean teacher that has been widely used in domain adaptive 2D object detection and LiDAR point cloud detection as a first baseline of unsupervised domain adaptation for indoor 3D object detection. Here we present more analysis on the results of such framework to illustrate its limitation and future direction of improving such framework.

As shown in Table 6, we test the models on the training set of target domain. In line "Oracle(target)", the performance is significantly higher than in the evaluation set due to the use of training set labels. In line "Source only" where the training set images and labels are unused, the results are similar because both sets are from the same distribution of target domain. However, in line "Mean teacher", despite using images from the target domain training set, the test results on the training set are not significantly higher than those on the evaluation set. We analyze the possible reasons as follows: 1) The pseudo-labels used in the mean teacher framework are of low quality, and misleading noisy pseudo-labels cause the model to not converge well on the training set; 2) The mean teacher framework relies solely on supervision from pseudo-labels and does not familiarize itself with the point cloud features of the target domain at the feature level. To address these two reasons, future improvements in domain adaptation work could focus on enhancing the model's training effectiveness under noisy conditions or introducing unsupervised methods to enable the model to familiarize itself with the target domain features without relying on labels.

A.5   EXPERIMENTS COMPUTE RESOURCES AND LIMITATIONS

We use VoteNet by Qi et al. (2019) as our base detector and conduct all the experiments on single NVIDIA A800 GPU with full precision training by PyTorch. Following Qi et al. (2019), we down-sample all the input scenes into 20,000 points, so the training time cost only rely on the scale of datasets. Under sour-only setting, training the ScanNet dataset with 1200 scenes takes approximately

Table 6: The evaluation results on training set and evaluation set of target domain

| | SUN RGB-D→ScanNet | | SimRoom→ScanNet | |
| | Training set | Evaluation set | Training set | Evaluation set |
|---|---|---|---|---|
| Source only | 31.26 | 28.85 | 5.67 | 5.25 |
| Mean teacher | 29.94 | 29.85 | 6.69 | 6.89 |
| Oracle(target) | 62.98 | 40.67 | 62.98 | 40.67 |

2 hours, while training the SUN RGB-D, SimRoom, and SimHouse datasets with 5000 to 6000 scenes takes about 10 hours. The source code and dataset files will be released soon.

Our work still has several limitations in the following aspects: 1) For controlling scene layout configuration, we only considered differences at the room level (number of rooms) and did not consider controlling more fine-grained differences at the instance level. For example, different indoor designers may place the furniture and object in different styles even in the same single room environment. In future work, we will use more sophisticated simulation data generators to achieve this control. 2) The generation of synthetic 3D environment is based on rules and the fixed set of 3D assets by Deitke et al. (2022) which is naturally unlike real-world 3D scenes. In the future we will introduce more powerful 3D generators such as diffusion-based neural networks to generate more realistic datasets.