# OpenReview forum: "Investigating Domain Gaps for Indoor 3D Object Detection"
_ICLR.cc/2025/Conference — Submitted to ICLR 2025_

### Official Review · Reviewer_hdju · 2024-10-19

**Soundness:** 2
**Presentation:** 2
**Contribution:** 2
**Rating:** 6
**Confidence:** 3

**Summary:**

This paper mainly discussed about indoor 3D object detection domain adaptation, which is an underexplored topic. It analyzes the statistics and characteristics of existing indoor datasets and proposes two synthetic datasets, SimRoom and SimHouse. This paper also conducted extensive experiments on different domain adaptation settings across datasets.

**Strengths:**

1. This paper focuses on an underexplored topic which is of potential.
2. The experiments are comprehensive.
3. This paper also proposes two large-scale synthetic datasets.

**Weaknesses:**

1. Lack of comparison with 3D-FRONT: A more thorough comparison between proposed SimRoom and SimHouse with 3D-FRONT [1] would be beneficial. 3D-FRONT is also of large quantity and high point cloud quality, with expert-designed room layouts which can potentially narrow the synthetic to real domain gap.

[1] Fu, Huan, et al. "3d-front: 3d furnished rooms with layouts and semantics." Proceedings of the IEEE/CVF International Conference on Computer Vision. 2021.

**Questions:**

1. I noticed that there was a recent work [1] which focuses on syn-to-real domain adaptation. A comparison with this paper would better evaluate the significance of the paper.
2. According to the results, the synthetic-to-real domain gaps seem difficult to narrow down without priors of the target dataset. Then is this setting practical?


[1] Wang, Yunsong, Na Zhao, and Gim Hee Lee. "Syn-to-Real Unsupervised Domain Adaptation for Indoor 3D Object Detection." arXiv preprint arXiv:2406.11311 (2024).

---

> ### Author Response · Authors · 2024-12-01
> **Response to Reviewer hdju**
>
> We thank you for your recognition that our paper explores a potential topic, conduct comprehensive experiments and proposes two large-scale synthetic datasets.
> We would like to response to your concerns in the following aspects:
>
> **1. The comparison of existing synthetic dataset 3D-Front.**
>
> We have already discussed the 3D Front dataset in the related work section of the main text and the supplementary material.
> In Section A.3 in the supplementary material, we analyze that our proposed SimRoom and SimHouse datasets which are
> generated by ProcTHOR framework have following advantages: (1) 3D-Front dataset is created by professional indoor designers with fixed layouts, thus with high cost and could neither be extended nor easily modified. Our proposed datasets is extensible and could scale up to any size with very low cost. (2) 3D-Front dataset contains limited number of categories and instances. Some object categories in our benchmark which are common among indoor scenes such as TV, toilet, and garbage can are not included or annotated in 3D-Front. In ProcTHOR framework we could add 3D models of any category and could easily configure their numbers and layouts. We train the detector with 3D-Front dataset as source domain on 8 categories (our proposed benchmark include 15 categories). As shown in Table 4 in the supplementary material, the synthetic to real adaptation (3D-Front to ScanNet) remains challenging. Additionally, domain gap exists between manually designed layout and automatically generated layout (3D-Front to SimHouse).
>
> **2. The comparison with more existing approaches.**
>
> As suggested by reviewers, we also tested OHDA[1], a method that had not been officially published at the time of our submission.
> This method combines synthetic-to-real data augmentation, the mean teacher framework, and adversarial training techniques. The performance of these methods and detailed analysis can be seen in Page 10, lines 517-528, and Table 3 of the revised version of our paper.
>
> **3. The practical meaning of exploring synthetic-to-real domain adaptation.**
>
> Real 3D point cloud data faces challenges such as difficulty in collection, annotation, and significant data differences, which have limited the scale of available datasets. Using synthetic data allows for the generation of large-scale datasets at a lower cost and faster speed, enabling the training of scalable 3D perception models. However, there is inevitably a domain gap between synthetic and real data, which affects the effectiveness of synthetic data in supporting real-world models. Therefore, synthetic-to-real adaptation has been a focal point in various domain adaptation studies. This paper focuses on investigating how to bridge different domain gaps, including the synthetic-to-real gap.
> Reviewer EsE4 describe our motivation as "well-articulated, effectively highlighting the necessity and significance of the research".
> Cross-domain evaluation gaps show that this is indeed a challenging topic, and we further study more detailed domain gaps, such as object distribution. We believe that future research in this area will enhance the domain adaptation capability of indoor point cloud object detectors, enabling the use of various types of data to train scalable models.

---

### Official Review · Reviewer_8Umj · 2024-11-01

**Soundness:** 3
**Presentation:** 3
**Contribution:** 2
**Rating:** 5
**Confidence:** 4

**Summary:**

The authors present the first benchmark for domain adaptation in indoor 3D object detection using existing standard datasets (ScanNet, SUN RGB-D) along with new synthetic datasets (SimRoom, SimHouse).

SimRoom and SimHouse are large-scale indoor point cloud datasets with a large number of objects and precise annotation.

These datasets are collected using a 3D simulator following the ProcTHOR.

With four datasets, the authors investigate four different domain adaptation scenarios: high-to-low-quality, low-to-high-quality, synthetic-to-real, and single-to-multiroom.

They evaluate various domain gap factors and observe that synthetic-to-real adaptation is the most challenging.

**Strengths:**

- The authors provide a thorough analysis of existing datasets, highlighting the differences between them.
Then, they clearly identify factors of challenging domain gaps across datasets.
Based on these investigations, they propose large-scale simulated indoor datasets (SimRoom, SimHouse) to complement existing datasets.

- The creation process of the large-scale dataset is comprehensively defined.
Through low-cost, precise annotation, they build a well-organized dataset.

- Through extensive experiments, the authors present meaningful benchmarks for various domain adaptation scenarios in indoor 3D object detection.

**Weaknesses:**

- Unlike datasets collected from real-world environments, synthetic datasets usually differ in texture detail and material realism.
Also, they have the risk of including simplistic object shapes and monotonous layouts.
These differences are likely significant factors in the large domain gap between synthetic-to-real datasets.
In this paper, the analysis of these differences is unclear, and it would be better to evaluate how well the proposed dataset reflects real-world conditions.

- While the proposed datasets include a large number of scene and object annotations, their significant domain gap from real-world datasets raises questions about their usefulness for various real-world settings.

- Although the dataset creation process is well-described, technical novelty is somewhat limited.
Their proposed framework, Mean Teacher, seems to simply follow previous work without adequately considering the diverse factors of domain gaps.

**Questions:**

- Did the authors consider experiments to observe domain gaps in the real-to-synthetic (scn2rm) settings?
These experiments could help support the applicability of the proposed dataset.

- How many different designs of each object are included in this synthetic dataset?
For example, in real-world environments, there is a variety of bed shapes or a diversity of chair designs.
Also, how many different types of room layouts are included?

- Did the authors consider resembling real-world environments when creating the scenes?
I have some concerns about the potential of this dataset to effectively complement existing datasets and be utilized in various 3D vision tasks.

While this work is well written, I have a few questions, as outlined in the weaknesses and questions. A clarification of the points I mentioned would help me improve my decision.

---

> ### Author Response · Authors · 2024-12-03
> **Response to Reviewer 8Umj (Part 1)**
>
> We thank you for your recognition that our paper analyzes the domain gaps of multiple indoor 3D scene datasets, introduces two large-scale synthetic datasets and proposes meaningful domain adaptation benchmarks.
> We would like to response to your concerns in the following aspects:
>
> **1. Detailed domain gap factors in synthetic to real setting should be discussed.**
>
> Thanks for your valuable suggestion. Synthetic to real setting contains fine-grained domain gap factors such as texture material and object  semantics.
> With the scalability and flexibility of ProcTHOR framework, we integrated objects from the ScanNet dataset to generate object placement layouts, with the objects being in an intermediate state derived from real-world data. As shown in the Table 4 in the revised paper, incorporating real objects leads to a noticeable performance improvement compared to fully simulated data, highlighting the semantic domain gap between datasets. However, compared to the target domain Oracle, the performance gap remains significant, suggesting that differences in object placement layouts are a more critical domain gap factor.
>
> **2. The practical meaning of proposing synthetic dataset and exploring synthetic-to-real domain adaptation.**
>
> Real 3D point cloud data faces challenges such as difficulty in collection, annotation, and significant data differences, which have limited the scale of available datasets. Using synthetic data allows for the generation of large-scale datasets at a lower cost and faster speed, enabling the training of scalable 3D perception models. However, there is inevitably a domain gap between synthetic and real data, which affects the effectiveness of synthetic data in supporting real-world models. Therefore, synthetic-to-real adaptation has been a focal point in various domain adaptation studies. This paper focuses on investigating how to bridge different domain gaps, including the synthetic-to-real gap. Reviewer EsE4 describe our motivation as "well-articulated, effectively highlighting the necessity and significance of the research". Cross-domain evaluation gaps show that this is indeed a challenging topic, and we further study more detailed domain gaps, such as object distribution. We believe that future research in this area will enhance the domain adaptation capability of indoor point cloud object detectors, enabling the use of various types of data to train scalable models.
>
> **3. The technical contribution for domain adaptation is limited.**
>
> As proposed in Section 4.3 of the main paper, we introduce multiple unsupervised domain adaptation techniques to improve performance on the target domain. When selecting domain adaptation methods, we considered the following approaches: (1) Mean teacher framework, which is commonly used in 2D domain adaptation and LiDAR point cloud domain adaptation; (2) VSS, proposed for domain-adaptive 3D semantic segmentation; (3) PPFA, proposed for semi-supervised learning on point clouds; (4) RV, designed for domain-adaptive point cloud classification.
> Furthermore, based on suggestions from other reviewers, we also tested OHDA[1], a method that had not been officially published at the time of our submission.
>
> Although we do not claim these approaches as our technical contributions, we re-implemented all of them, making adjustments for the indoor 3D object detection task and reporting the improved results for a comprehensive evaluation.
>
> [1] Yunsong Wang, Na Zhao, and Gim Hee Lee. Syn-to-Real Unsupervised Domain Adaptation for Indoor 3D Object Detection. arXiv preprint arXiv:2406.11311 (2024).

---

> ### Author Response · Authors · 2024-12-03
> **Response to Reviewer 8Umj (Part 2)**
>
> **4. Suggesting to conduct experiments on real to synthetic setting.**
>
> The original intention of designing the synthetic-to-real domain adaptation scenario is to use large-scale and diverse synthetic data to assist in understanding real-world scenes, addressing the issue of insufficient real-world data. However, when the synthetic world is treated as the target domain, there is already a sufficiently diverse and ample amount of in-distribution data available for model training. Therefore, we did not include the real-to-synthetic setting in our benchmark.
>
> | Method   | Source only | VSS  | MT | OHDA | RV | Oracle (target) |
> |:----------:|:----------:|:----------:|:----------:|:----------:|:----------:|:----------:|
> | scan2room  | 27.01  |  24.33  | 31.42 | 31.75 | 32.42 | 76.73 |
>
> Thank you for your suggestion. We conducted experiments in the real-to-synthetic setting, and the results are shown in the table above. The results indicate that models trained with small-scale, low-diversity, and out-of-distribution real data and labels have a significant performance gap compared to models trained with in-distribution synthetic data. Since synthetic data and labels are easily accessible, directly using synthetic data for training would be the optimal choice for improving model performance in the synthetic world.
>
> **5. How many unique designs of objects and layouts are included in our proposed synthetic dataset?**
>
> We used the ProcTHOR framework to generate synthetic data. This framework supports all 3D object assets provided by the AI2-THOR system, with a total of 3,578 unique 3D assets (excluding room structural components such as walls, doors, windows, floors, and ceilings) across 108 categories. Furthermore, the framework supports random selection of object materials, colors, and scene lighting during the generation process. Therefore, when considering the color attributes of each point in the point cloud, objects with the same shape can appear as different point clouds in different generated scenes. As shown in Table 1 of the main text, our proposed dataset includes bounding box annotations for millions of objects.
>
> For scene layout generation, the ProcTHOR framework uses random seed values to ensure that each scene has a different layout. As a result, our proposed SimRoom and SimHouse datasets have 7,202 and 7,306 unique layouts, respectively.
>
> Additionally, due to the flexibility of the generation framework, we are able to replace 3D object assets from any source during the generation process, and we can also introduce other layouts to simulate the placement of objects. We mentioned the experiment of replacing ScanNet objects in response of question 1.
>
> **6. Suggesting to resemble real-world environments when creating the scenes**
>
> We mentioned the experiment of replacing ScanNet objects in response of question 1.

---

### Official Review · Reviewer_EsE4 · 2024-11-03

**Soundness:** 3
**Presentation:** 3
**Contribution:** 3
**Rating:** 6
**Confidence:** 4

**Summary:**

This study introduces novel benchmarks for domain adaptation in indoor 3D object detection, utilizing established datasets (ScanNet and SUN RGB-D) alongside two newly created large-scale synthetic datasets, SimRoom and SimHouse, which contain a greater number of objects and provide highly accurate annotations.The authors systematically examine various domain gaps, including point cloud quality, differences between synthetic and real-world data, room layout, and object size. Cross-dataset experiments reveal that adapting from synthetic to real-world data poses the most significant challenge. To address these challenges, the study evaluates several domain adaptation techniques as baseline methods to improve model performance in cross-domain scenarios. This work lays a foundation for future research aimed at developing robust indoor 3D object detectors capable of generalizing across diverse indoor environments.

**Strengths:**

This paper focuses on indoor 3D object detection, moving beyond the traditional emphasis on outdoor environments. The authors introduce two larger, novel synthetic datasets to facilitate exploration in this area. Through experiments conducted across multiple adaptation scenarios, they analyze critical factors such as point cloud quality and object size, thoroughly investigating their impact on model adaptability.The structure of the paper is clear, with a rigorous logical flow and fluent expression. The motivation for the study is well-articulated, effectively highlighting the necessity and significance of the research. These contributions provide a substantial foundation for further investigation into domain adaptation in indoor 3D object detection, showcasing notable academic value and practical application potential

**Weaknesses:**

1.The authors emphasize the low annotation costs associated with their synthetic datasets. However, a comparative analysis of the annotation costs, speeds, and methodologies between their simulation data generation approach and other synthetic data generation techniques, as well as traditional manual annotation methods, would strengthen their argument. Including such an analysis could provide insights into the efficiency and scalability of their approach relative to existing methods, highlighting the practical advantages or limitations of using simulated data.
2.The paper would benefit from a detailed comparison with recent works. For example, the unsupervised domain adaptation method for indoor 3D detection by Wang et al. ([1]) presents an alternative approach that could be discussed alongside the authors' findings. Highlighting the similarities, differences, and potential synergies between these methods would provide valuable context for readers.
Additionally, Lu et al.'s work ([2]), which focuses on open-vocabulary 3D object detection without the need for 3D annotations, offers useful benchmarks. The authors could reference Table 3 from that study, which examines transfer learning effectiveness in indoor 3D detection. Comparing their results with those of Lu et al. would clarify how their methods contribute to the state of the art and identify areas for further exploration.
[1] Yunsong Wang. Syn-to-Real Unsupervised Domain Adaptation for Indoor 3D Object Detection, BMVC 2024
[2] Yuheng Lu. Open-Vocabulary Point-Cloud Object Detection without 3D Annotation, CVPR 2023

**Questions:**

Please refer to the weaknesses section above.

---

> ### Author Response · Authors · 2024-12-02
> **Response to Reviewer EsE4 (part 1)**
>
> We thank you for your recognition that our paper has a well-articulated motivation, introduces large-scale datasets, analyzes critical factors for domain gaps of indoor scenes, and has a rigorous logical flow and fluent expression.
> We would like to response to your concerns in the following aspects:
>
> **1. The efficiency analysis of dataset construction and annotation.**
>
> We propose using the ProcTHOR framework for low-cost, flexible, and scalable generation of 3D indoor scenes.
> Previous indoor 3D datasets, whether designer-built synthetic datasets or real room scan datasets, rarely mention the time and financial costs involved in their creation.
> The 3D-Front [1] dataset from the simulated environment only discloses that it is derived from 5,000 indoor rooms designed by expert designers in the company’s historical database, and that the dataset's organization and refinement process also relied heavily on the work of designers.
> The real-world ScanNet [2] dataset mentioned that they use a commodity RGB-D sensor attached to handheld devices such as an iPhone or iPad to scan thousands of rooms.
> It has annotations by more than 500 crowd workers on the Mechanical Turk platform.
> While the time and financial cost in these datasets are not explicitly stated, it is clear that the costs are considerable.
>
> Our proposed dataset is fully automated using the ProcTHOR framework, where the generation process determines the precise absolute positions of objects within the scene, eliminating the need for additional efforts in annotation. Based on our measurements from running the generation framework locally on a computer with an AMD Ryzen 7 5700g CPU and 32GB of RAM, the time to generate a single house scene configuration is approximately 13.7 seconds. Using Unity to render it into a .obj format mesh file takes about 64.3 seconds, and sampling point clouds from the mesh file takes around 1.5 seconds. This generation efficiency far exceeds that of other synthetic datasets that rely on human involvement or real-world datasets.
>
> After the review results are announced, we will open-source all our code, including the generation code for SimRoom and SimHouse (allowing users to generate simulation datasets of any scale) as well as all baseline domain adaptation methods we implemented on the benchmarks. At this time, we provide a part of our proposed datasets here. Through our work, we hope to inspire more future research on scene-level domain adaptation for indoor point cloud data.
>
> [1] Fu, Huan, et al. "3d-front: 3d furnished rooms with layouts and semantics." ICCV 2021.
>
> [2] Dai, Angela, et al. "Scannet: Richly-annotated 3d reconstructions of indoor scenes." CVPR 2017.

---

> ### Author Response · Authors · 2024-12-02
> **Response to Reviewer EsE4 (part 2)**
>
> **2. The comparison with more existing approaches.**
>
> As suggested by reviewers, we also tested OHDA[1], a method that had not been officially published at the time of our submission. This method combines synthetic-to-real data augmentation, the mean teacher framework, and adversarial training techniques. The performance of these methods and detailed analysis can be seen in Page 10, lines 517-528, and Table 3 of the revised version of our paper.
>
> As for OV-3DET [2], thank you for your suggestion, and we have reviewed several open vocabulary methods for indoor 3D object detection and instance segmentation, many of which report cross-domain testing performance.
> We evaluate [2] on part of our proposed benchmarks, and report the results in the following table.
>
> | Method | scan2sun | sun2scan |
> |--------------------|--------------------|--------------------|
> | VoteNet source only | 20.96 | 33.56 |
> | VoteNet target oracle | 37.52 | 44.09 |
> | OV-3DET source only | 17.37 | 10.22 |
> | OV-3DET target oracle | 20.79 | 27.92 |
>
> However, we believe these methods are outside the scope of comparison in this paper for the following two reasons:
>
> 1. OV-3DET uses a backbone network different from that of fully supervised object detection, and due to the lack of annotations, its baseline performance lags far behind that of fully supervised methods (target oracle 37.52 vs 20.79, 44.09 vs 27.92). We believe that a major bottleneck in addressing the cross-domain performance gap for unsupervised or open vocabulary problems lies in the cross-domain generalization ability of the 3D detector itself. Therefore, this paper focuses on cross-domain evaluation of the basic 3D object detector, without directly comparing semi-supervised, unsupervised, or open vocabulary methods.
>
> 2. [2] relies on 2D images paired with scene point clouds and the camera intrinsic and extrinsic parameters used to capture them, utilizing additional 2D information as auxiliary data to enable open vocabulary detection. In the benchmark we propose, the 3D point cloud datasets constructed from RGB-D images or videos contain this information, while point clouds directly sampled from synthetic 3D meshes lack inherent camera positions and parameters. Therefore, in the table above, we report only the results for the scan2sun and sun2scan benchmarks. This paper focuses on cross-domain evaluation of pure 3D point clouds, and the methods implemented in our main paper do not rely on additional 2D information. Using 2D information to assist cross-domain scene understanding will be part of our future work.
>
> [1] Yunsong Wang. Syn-to-Real Unsupervised Domain Adaptation for Indoor 3D Object Detection, BMVC 2024
>
> [2] Yuheng Lu. Open-Vocabulary Point-Cloud Object Detection without 3D Annotation, CVPR 2023

---

### Official Review · Reviewer_8mjq · 2024-11-03

**Soundness:** 2
**Presentation:** 2
**Contribution:** 2
**Rating:** 3
**Confidence:** 4

**Summary:**

This paper explores the task of domain adaptation in the task of indoor 3D object detection by building a new domain adaptation benchmark. The benchmark datasets include the widely-used ScanNet and SUN RGB-D datasets, as well as the newly proposed SimRoom and SimHouse synthetic datasets. SimHouse dataset is generated by sampling large-scale house layouts with ProcTHOR framework originally designed for embodied AI tasks, and SimRoom is generated by splitting SimHouse scenario into different rooms. Four domain adaptation settings are explored, and the experiments show that the synthetic to real domain gap is the most significant domain gap. Extensive experiments are conducted on this benchmark, providing the first baseline for the domain adaptation of indoor 3D object detection.

**Strengths:**

1. The paper generated a large-scale synthetic dataset for ablating indoor 3D object detection task, which may help multiple potential tasks for indoor scene understanding.
2. The paper demonstrates extensive experiments to ablate different basic approaches on the newly proposed domain adaptation benchmark.

**Weaknesses:**

1. This paper does not provide adequate discussions about the applications and importance of the task of domain adaptation in indoor 3D object detection. Moreover, it also doesn't discuss the differences and unique values compared with the task of outdoor domain adaptation of 3D object detection.
2. It seems that the generation process of the SimRoom / SimHouse dataset is a simple usage of ProcTHOR framework.
3. For the domain adaptation benchmark, it is also the application of multiple existing approaches, and the technical contribution is limited.

**Questions:**

1. The authors claim that this is the first baseline for domain adaptive indoor 3D object detection. However, these already exist many papers about domain adaptive outdoor 3D object detection. Hence, it is necessary to compare with domain adaptive outdoor 3D object detection task and highlight the unique values / differences between these two tasks.
2. Typos:
a) L093: SUN RGB-D -> SUN RGB-D?
b) L262: static -> statistic
c) L320: out of the scope

---

> ### Author Response · Authors · 2024-12-01
> **Response to Reviewer 8mjq**
>
> We thank you for your recognition that our paper proposes large-scale indoor 3D object detection datasets, and conducts extensive experiments to analysis the domain gaps for indoor point clouds.
> We would like to response to your concerns in the following aspects:
>
> **1. The lack of discussion on the unique values / differences between domain adaptive indoor 3D object detection and domain adaptive LiDAR-based outdoor 3D object detection.**
>
> In the related work section of the main text (Page 6, lines 186-200), we discuss domain adaptation methods for LiDAR point cloud detection and highlight the unique challenges of this task in indoor point clouds.
> We would like to further explain the unique differences in the following aspects:
>
> - **Domain gap factors**:
> Outdoor point cloud datasets are collected from LiDAR in autonomous driving street scenes, featuring relatively similar landscapes and uniform point distribution patterns. Research on their domain adaptation mainly focuses on differences in car characteristics, LiDAR scanning beams, and weather conditions.
> [1] first studies domain gaps for outdoor point cloud detection, conducting experiments on large-scale autonomous driving datasets such as nuScenes, Waymo, and KITTI, and concluding that the primary adaptation hurdle to overcome is the differences in car sizes across geographic areas.
> [2] proposes a method for adaptation between different LiDAR scan beams.
> [3] proposes bridging the weather domain gap for LiDAR detectors.
> Subsequent work has been conducted on the above benchmark, designing general methods to address the aforementioned domain gaps.
> Our paper is the first comprehensive domain adaptation benchmark for indoor point cloud detection, investigating domain gaps such as point scanning quality (with or without significant point loss), synthetic vs. real data (both layout and object semantic gaps), and room layout configuration (single rooms vs. houses).
>
> - **Object categories**: Outdoor domain adaptation focuses only on cars ([1, 2, 3] evaluate results only for the car category), while indoor object detection requires identifying and locating many more object categories (15 in our benchmarks) with rich semantic information.
>
> - **Object location distribution**: The distance between cars or other objects is relatively large in outdoor road scenes. As a result, BEV methods can be applied for detector architecture, and point completion-based methods could be used [4] for domain adaptation.
> However, indoor scenes feature closely placed objects, with even overlapping bounding boxes.
>
> Overall, outdoor LiDAR point cloud detection and indoor point cloud detection belong to two distinct domains, using different detector architectures and evaluation standards. Moreover, the scale of indoor point cloud datasets is much smaller than that of autonomous driving data, which further increases the difficulty.
>
> [1] Wang, Yan, et al. "Train in germany, test in the usa: Making 3d object detectors generalize." CVPR, 2020.
>
> [2] Wei, Yi, et al. "Lidar distillation: Bridging the beam-induced domain gap for 3d object detection." ECCV, 2022.
>
> [3] Hahner, Martin, et al. "Lidar snowfall simulation for robust 3d object detection." CVPR, 2022.
>
> [4] Xu, Qiangeng, et al. "Spg: Unsupervised domain adaptation for 3d object detection via semantic point generation." ICCV 2021.
>
> **2. The usage of ProcTHOR framework.**
>
> The focus of this paper is to study the domain gap factors in indoor 3D point clouds.
> We use the ProcTHOR framework to generate large-scale simulated indoor scene datasets, without conducting innovative research on the generation process itself.
> We adopted this framework for its scalability and flexibility.
> With these features, we are able to control the variables of various domain gap factors and investigate domain adaptation tasks in 3D point cloud detection.
> Leveraging the flexibility of the generative data framework, we integrated objects from the ScanNet dataset to generate object placement layouts, with the objects being in an intermediate state derived from real-world data.
> As shown in the Table 4 in the revised paper, incorporating real objects leads to a noticeable performance improvement compared to fully simulated data, highlighting the semantic domain gap between datasets. However, compared to the target domain Oracle, the performance gap remains significant, suggesting that differences in object placement layouts are a more critical domain gap factor.
>
> **3. The technical contribution for domain adaptation is limited.**
>
> We evaluate existing domain adaptation approaches on our proposed benchmarks.
> Although we do not claim these approaches as our technical contributions, we re-implemented all of them, making adjustments for the indoor 3D object detection task and reporting the improved results for a comprehensive evaluation.

---

### Official Review · Reviewer_wt3L · 2024-11-04

**Soundness:** 2
**Presentation:** 3
**Contribution:** 2
**Rating:** 5
**Confidence:** 3

**Summary:**

The paper investigates the challenge of domain adaptation for indoor 3D object detection by introducing new synthetic datasets, SimRoom and SimHouse, and presenting domain adaptation benchmarks across several main scenarios. The paper examines and analyzes factors such as point cloud quality, room layout, synthetic-to-real style, and object size to highlight the issue of domain adaptation for indoor 3D scenes. Additionally, the authors provide baseline results using simple adaptation methods to assess the performance of models trained on one domain and tested on another.

**Strengths:**

The paper provides a detailed analysis of the impact of domain gaps on model performance. This is useful for understanding specific adaptation challenges and will benefit future work to address these potential issues in real practice.
Besides showing the challenges, the proposed synthetic datasets, SimRoom and SimHouse, also offer more diverse and scalable data compared to existing real-world datasets, enabling controlled experiments for domain adaptation.
Overall, this paper presents a novel benchmark specifically for indoor 3D object detection domain adaptation, contributing to the field's progress.

**Weaknesses:**

The baseline domain adaptation methods implemented are straightforward and lack complexity. Methods like the mean teacher framework and size priors are standard and do not demonstrate significant innovation or exploration of recent advancements in domain adaptation, using such basic adaptation methods might be insufficient to challenge future models.
In addition, although the paper claims that the synthetic datasets have high-quality annotations, there is a lack of discussion about how faithfully these datasets mimic real-world scenarios. More detailed analysis of data consistency and potential annotation noise could demonstrate the reliability and generalizability of the results.
Also, the paper could provide more information about the consistency and quality assurance processes used in annotating SimRoom and SimHouse. Since the study utilizes the SimRoom and SimHouse with our existing datasets, inconsistencies or errors in labeling between these datasets collected can lead to unreliable benchmarks and misleading conclusions. Further discussion demonstrating the consistency of the data collection and labeling processes for these datasets will be beneficial for establishing the benchmark.

**Questions:**

It'll be helpful if the authors can provide more details on how data quality and annotation consistency were ensured in the SimRoom and SimHouse datasets.
I hope the authors provide more details about why they chose to implement baseline methods such as few-shot fine-tuning or mean teacher framework for domain adaptation instead of considering more methods, such as adversarial training or self-supervised learning, for stronger comparisons.
Most importantly, I was wondering if the authors have any plans to make SimRoom and SimHouse publicly available.
I appreciate any responses to these questions!

---

> ### Author Response · Authors · 2024-12-01
> **Response to Reviewer wt3L**
>
> We thank you for your recognition that our paper proposes a novel and meaningful benchmark and provides synthetic datasets with more diverse and scalable data.
> We would like to respond to your concerns in the following aspects:
>
> **1. Suggesting to evaluate more complex domain adaptation techniques such as adversarial training and self-supervised learning.**
>
> As proposed in Section 4.3 of the main paper, we introduce multiple unsupervised domain adaptation techniques to improve performance on the target domain.
> When selecting domain adaptation methods, we considered the following approaches:
> (1) Mean teacher framework, which is commonly used in 2D domain adaptation and LiDAR point cloud domain adaptation;
> (2) VSS, proposed for domain-adaptive 3D semantic segmentation;
> (3) PPFA, proposed for semi-supervised learning on point clouds;
> (4) RV, designed for domain-adaptive point cloud classification.
>
> Specifically, PPFA utilizes adversarial training as part of its approach, and RV incorporates self-supervised learning when refining pseudo labels.
> Furthermore, based on suggestions from other reviewers, we also tested OHDA[1], a method that had not been officially published at the time of our submission.
> This method combines synthetic-to-real data augmentation, the mean teacher framework, and adversarial training techniques.
> The performance of these methods and detailed analysis can be seen in Page 10, lines 517-528, and Table 3 of the revised version of our paper.
> Note that although we do not claim these approaches as our technical contributions, we re-implemented all of them, making adjustments for the indoor 3D object detection task and reporting the improved results for a comprehensive evaluation.
> We will release the details of these re-implemented methods in our open-source code after the review process.
>
> [1] Yunsong Wang, Na Zhao, and Gim Hee Lee. Syn-to-Real Unsupervised Domain Adaptation for Indoor 3D Object Detection. arXiv preprint arXiv:2406.11311 (2024).
>
> **2. Deteiled explanation is needed on how the synthetic framework mimics real-world point clouds, as well as how data quality and annotation consistency are ensured.**
>
> As mentioned in Section 3.2 of the main paper, we use the ProcTHOR[1] framework to generate synthetic 3D house configurations and sample point clouds from the 3D meshes.
>
> ProcTHOR mimics real-world house configurations by utilizing a combination of realistic 3D models and algorithms that generate indoor environments with physical and semantic properties similar to those found in real homes.
> It simulates realistic object interactions and physical properties, such as object occlusion, lighting, and shadows, to make the generated environments more realistic and interactive.
> Though specially designed for mimicking the real world, the domain gap in synthetic data still exists.
> We adopted this framework for its scalability and flexibility. With these features, we are able to control the variables of various domain gap factors and investigate domain adaptation tasks in 3D point cloud detection.
>
> The generation process involves first sampling the layout of the room's floor and walls, followed by a progressive sampling process to place large objects, wall objects, and small surface objects at specific locations within the room.
> The generated results are combinations of objects with determined relative positions in the simulator. As a result, the generated rooms inherently define the exact position of each object without the need for additional manual or automated labeling.
> Compared to real-world scene point clouds, which rely on manual annotation, this method is more cost-effective and provides inherently accurate ground truth for object detection.
>
> [2] Matt Deitke, Eli VanderBilt, Alvaro Herrasti, Luca Weihs, Kiana Ehsani, Jordi Salvador, Winson Han, Eric Kolve, Aniruddha Kembhavi, and Roozbeh Mottaghi. Procthor: Large-scale embodied ai using procedural generation. Advances in Neural Information Processing Systems, 35:5982–5994, 2022.
>
> **3. Future plans of open-source project.**
>
> After the review results are announced, we will open-source all our code, including the generation code for SimRoom and SimHouse (allowing users to generate simulation datasets of any scale) as well as all baseline domain adaptation methods we implemented on the benchmarks.
> At this time, we provide a part of our proposed datasets [here](https://drive.google.com/file/d/1vKn6V7RMUhMuVut7hAND-g3VaOl2kwEA/view?usp=drive_link).
> Through our work, we hope to inspire more future research on scene-level domain adaptation for indoor point cloud data.

---

### Author Response · Authors · 2024-12-01
**General official comment**

We thank all the reviewers for their careful reviews and constructive suggestions. We also appreciate your recognition that our paper:
- proposes a novel and meaningful benchmark with well-articulated motivation (Reviewer wt3L, Reviewer EsE4, Reviewer hdju)
- proposes valuable datasets with diverse and scalable data (Reviewer wt3L, Reviewer 8mjq, Reviewer EsE4, Reviewer 8Umj, Reviewer hdju)
- conducts extensive experiments to analyze the domain gaps for indoor point clouds (Reviewer wt3L, Reviewer 8mjq, Reviewer EsE4, Reviewer 8Umj, Reviewer hdju)
- is clearly written with a rigorous logical flow and fluent expression (Reviewer EsE4)

Based on the suggestions of the reviewers, we have submitted a revised version of the paper, with the changes highlighted in blue. The revisions specifically include the following aspects:
- We emphasized the advantages of using the framework to generate synthetic datasets and highlighted further experiments we conducted by leveraging the flexibility of the generation framework (Page 5, lines 256-262).
- We further explained the domain gap differences between different datasets, which include not only the global average size differences but also the semantic differences of the objects (Page 6, lines 315-323).
- We added experiments where the generation framework was used to apply real scene object data to generate simulated layouts, producing a domain that lies between synthetic and real data. We analyzed the impact of object semantics and the layout of objects within rooms on domain adaptation performance (Page 8, lines 426-431 and Table 4).
- We revised the textual description of the performance of different domain adaptation methods on our benchmark, added the domain adaptation methods suggested by the reviewers, and emphasized the implementation details of each method (Page 10, lines 517-528 and Table 3).

We would like to once again thank the reviewers for their re-evaluation. We will address each reviewer's concerns one by one below.

---

### Meta-Review · Area_Chair_HF8r · 2024-12-20

**Metareview:**

This paper investigates domain adaptation for indoor 3D object detection by introducing new synthetic datasets, SimRoom and SimHouse, and providing domain adaptation benchmarks across four main scenarios: real-to-real (with different point cloud qualities), synthetic-to-real, and synthetic single-room to multi-room. The key contributions include the creation of synthetic room-level and house-level datasets using ProcThor, along with the design of an indoor 3D object detection domain adaptation benchmark and evaluation of existing unsupervised domain adaptation (UDA) techniques.

The paper received mixed reviews, with two reviewers (EsE4, hdju) voting for borderline acceptance and three reviewers (wt3L, 8mjq, 8Umj) voting for rejection.

The AC noted that none of the reviewers responded to the authors' rebuttal. After reviewing the authors' responses, the AC believes that some of the concerns were addressed, including the suggestion to evaluate more complex domain adaptation techniques (wt3L) and the request to discuss the differences between domain adaptation for indoor 3D object detection and LiDAR-based outdoor 3D object detection (8mjq). However, several key concerns remain, such as:

- **Feasibility of resembling real-world environments** when creating scenes to complement existing datasets (8Umj). The integration of ScanNet objects does not fully address this issue.

- **Concerns about the SimRoom/SimHouse dataset generation process**, which is seen as a simple usage of the ProcThor framework (8mjq), without ensuring annotation consistency or mitigating noise between synthetic and real datasets (wt3L).

- **Lack of discussion on the practicality of synthetic-to-real domain adaptation**, especially when considering that cross-dataset adaptation (e.g., SUN→Scan) performs much better than synthetic-to-real adaptation (e.g., Room→Scan) (hdju).

While the AC appreciates the authors’ efforts in creating a benchmark for indoor 3D object detection domain adaptation, the aforementioned concerns significantly outweigh the contributions. Therefore, the AC recommends rejecting this paper.

**Additional Comments On Reviewer Discussion:**

The paper received mixed reviews, with two reviewers (EsE4, hdju) voting for borderline acceptance and three reviewers (wt3L, 8mjq, 8Umj) voting for rejection.

The AC noted that none of the reviewers responded to the authors' rebuttal. After reviewing the authors' responses, the AC believes that some of the concerns were addressed, including the suggestion to evaluate more complex domain adaptation techniques (wt3L) and the request to discuss the differences between domain adaptation for indoor 3D object detection and LiDAR-based outdoor 3D object detection (8mjq). However, several key concerns remain, such as:

- **Feasibility of resembling real-world environments** when creating scenes to complement existing datasets (8Umj). The integration of ScanNet objects does not fully address this issue.

- **Concerns about the SimRoom/SimHouse dataset generation process**, which is seen as a simple usage of the ProcThor framework (8mjq), without ensuring annotation consistency or mitigating noise between synthetic and real datasets (wt3L).

- **Lack of discussion on the practicality of synthetic-to-real domain adaptation**, especially when considering that cross-dataset adaptation (e.g., SUN→Scan) performs much better than synthetic-to-real adaptation (e.g., Room→Scan) (hdju).

While the AC appreciates the authors’ efforts in creating a benchmark for indoor 3D object detection domain adaptation, the aforementioned concerns significantly outweigh the contributions. Therefore, the AC recommends rejecting this paper.

---

### Decision · Program_Chairs · 2025-01-22

Reject